# Infection increases activity via Toll dependent and independent mechanisms in *Drosophila melanogaster*

Crystal M. Vincent[1,2☯], Esteban J. Beckwith[1,2☯¤], Carolina J. Simoes da Silva[1,2], William H. Pearson[1,2], Katrin Kierdorf[3,4,5], Giorgio F. Gilestro[2], Marc S. Dionne[1,2]*

1 MRC Centre for Molecular Bacteriology and Infection, Imperial College London, London, United Kingdom, 2 Department of Life Sciences, Imperial College London, London, United Kingdom, 3 Institute of Neuropathology, Faculty of Medicine, University of Freiburg, Freiburg, Germany, 4 Center for Basics in NeuroModulation (NeuroModulBasics), Faculty of Medicine, University of Freiburg, Freiburg, Germany, 5 CIBSS-Centre for Integrative Biological Signalling Studies, University of Freiburg, Freiburg, Germany

☯ These authors contributed equally to this work.
¤ Current address: Instituto de Fisiología, Biología Molecular y Neurociencias (IFIBYNE), UBA-CONICET, Buenos Aires, Argentina
* m.dionne@imperial.ac.uk

**Data Availability Statement:** All relevant data are available at Zenodo: https://zenodo.org/record/ 6567039 (DOI 10.5281/zenodo.6567039). All other

## Abstract

Host behavioural changes are among the most apparent effects of infection. 'Sickness behaviour' can involve a variety of symptoms, including anorexia, depression, and changed activity levels. Here, using a real-time tracking and behavioural profiling platform, we show that in *Drosophila melanogaster*, several systemic bacterial infections cause significant increases in physical activity, and that the extent of this activity increase is a predictor of survival time in some lethal infections. Using multiple bacteria and *D. melanogaster* immune and activity mutants, we show that increased activity is driven by at least two different mechanisms. Increased activity after infection with *Micrococcus luteus*, a Gram-positive bacterium rapidly cleared by the immune response, strictly requires the *Toll* ligand *spätzle*. In contrast, increased activity after infection with *Francisella novicida*, a Gram-negative bacterium that cannot be cleared by the immune response, is entirely independent of both Toll and the parallel IMD pathway. The existence of multiple signalling mechanisms by which bacterial infections drive increases in physical activity implies that this effect may be an important aspect of the host response.

## Non-technical author summary

Sickness behaviours are often observed during infection. Animals have been shown to change their feeding, mating, social and resting (sleeping) behaviours in response to infection. We show here that fruit-flies infected with bacteria respond by increasing their physical activity and decreasing the amount of time spent sleeping. This increase in activity is seen in some, but not all, bacterial infections, and appears to be driven by at least two different mechanisms: with some bacteria, activating the immune response is the only

data are within the manuscript and its Supporting information files.

**Funding:** This work was supported by the Medical Research Council (MR/R00997X/1 to CMV and MSD; MR/L018802/2 to MSD), the Wellcome Trust (207467/Z/17/Z to MSD), the Biotechnology and Biological Sciences Research Council (BB/R018839/1 to GFG; BB/L020122/2 to MSD), the Deutsche Forschungsgemeinschaft (to KK), the European Commission (705930 to EJB), and the International Brain Research Organization (to EJB). The funders had no role in study design, data collection and analysis, decision to publish, or preparation of the manuscript.

**Competing interests:** The authors have declared that no competing interests exist.

requirement to induce increased activity, while other bacteria induce increased activity independently of known immune detection pathways. The biological role of increased activity is unclear; flies in the wild may be driven to flee sites where infection risk or pathogen burden is high. Alternatively, increased activity could serve a less direct anti-microbial function. For example, active animals may be more likely to encounter potential mates or food resource.

## Introduction

Some of the most apparent effects of infection are the sickness behaviours of the host. A variety of infection-induced behavioural changes have been documented; in vertebrates, these commonly include anorexia, lethargy, and social withdrawal [1–3]. In insects, a partially overlapping set of changes have been described, including anorexia and foraging changes, behavioural fevers, and changes in oviposition [4–8]. These changes in behaviour can facilitate immune function, either in terms of pathogen clearance or host survival; in some instances, the pathogen appears to benefit; while in others, there is no obvious beneficiary, in which case, the observed behavioural change may be a non-selected consequence of the interaction of two or more complex physiological systems. In all these instances, infection behaviours have a strong effect on the well-being of the host, irrespective of whether the effect is ultimately manifested as a difference in infection outcome.

Whilst sickness behaviours are often described as being part of the host response to infection, several studies have shown that behavioural changes during infection can also be the result of pathogen manipulation of host biology, rather than the host responding to a pathogen threat, *per se* [9–13]. The difference between a host response and parasite manipulation is not simply a matter of semantics, as the two can predict opposing evolutionary trajectories and infection outcomes [12,14,15]. If hosts change their behaviour in response to the physiological stresses associated with infection, we assume that said behaviour will be of benefit to the host, usually by reducing pathology [16–19]. In contrast, when pathogens manipulate host behaviour, we assume it serves the function of increasing pathogen fitness, often via enhanced transmission [10,20–25].

Changes in sleep and activity are some of the most common behavioural manifestations of infection, seen in vertebrates and invertebrates [26–28]. The extensive crosstalk between sleep and immunity has led to many suppositions regarding the value of sleep in maintaining a robust immune response and health in the face of infection [26,27,29–31]. However, despite investigations into the interplay between sleep and infection in insects, there remain inconsistencies as to whether sleep (or activity) is induced or inhibited during infection, and what effect these changes have on infection pathology [17,27,28,32–34]. Whilst some of this incongruity may result from the fact that in these studies flies were injected at different times of the day [35], they could also be caused by differences between pathogens used and therefore disparate activation of immune factors [36–40]. The consequences of infection-induced changes in sleep and activity are thus multifaceted and the effects of infection will depend on interactions between host immune and nervous systems, as well as the pathogen itself.

Here, using the real-time tracking and behavioural profiling platform, the ethoscope [41], we test an array of various bacteria and *D. melanogaster* immune and activity mutants, to determine whether pathogen recognition and immune pathway activation contribute to the increase in activity that we observe during infection.

## Results

### Bacterial infection leads to a marked increase in locomotor activity

We began by exploring the effects of *Francisella novicida* on physical activity in *Drosophila melanogaster*; a Gram-negative bacterium that propagates both intra- and extra-cellularly in *D. melanogaster* [42,43]. This infection is particularly tractable for behavioural studies because it presents near-synchronous mortality; an infection course in excess of three days, allowing ample time for activity monitoring; and strong immune activation, allowing for the identification of effects of immune activation on activity [42,44]. We found that male flies infected with *F. novicida* spent significantly more time moving than mock injected and uninfected controls (Fig 1A). This increase in activity intensified over the course of infection and was not observed in females (S1A Fig). Further partitioning of the activity data found that while there were subtle increases in micro-movements such as grooming and feeding [41,45] (Fig 1B and S2A Fig), the observed increase in movement was primarily the result of increased time spent walking (Fig 1C and S2B Fig); correspondingly, infected flies spent less time sleeping than mock injected and uninfected controls (Fig 1D and S2C Fig). In looking at the total distance travelled by infected flies relative to the total time they spent active, we found no difference between infected and uninfected animals, indicating that the intensity of their activity was unaltered by infection (S2D and S2E Fig). Furthermore, the increase in activity observed in infected flies was maintained when we considered each day of infection separately (S3A Fig). Activity on the first day of infection was predictive of lifespan, with more active flies exhibiting increased survival (Fig 1E). While there was a positive correlation between total activity and survival (S3B Fig), we used activity on day one as a predictor because previous work has found immune activity within the first few hours of infection to be both potent and an important determinant of survival [37,46–49]. Importantly, day one activity levels were positively correlated with total activity levels (S3C Fig), giving us confidence that activity on day one is representative of total activity levels in assessing infections of longer duration.

To test whether greater activity following infection was specific to *F. novicida* infection or a general consequence of immune activation, we infected wild-type flies with a phylogenetically and pathogenically diverse panel of bacteria and monitored activity either until all flies had died as a result of infection, or for the first four days after injection. We found that three of the five bacteria examined, *Micrococcus luteus*, *Listeria monocytogenes*, and *Staphylococcus aureus*, induced increased activity (Fig 2A–2C and S4 Fig). As observed during infection with *F. novicida*, increased activity was correlated with increased survival in flies infected with *L. monocytogenes*, however, this correlation was not seen in flies infected with *S. aureus* (Fig 2D–2E). Next, we screened a *D. melanogaster* wild-type (Oregon-R), as well as a selection of immune, locomotor and circadian mutants for activity levels during *F. novicida* infection and observed increased locomotor activity in all lines tested (S5 Fig and S1 and S2 Tables). The fact that many—but not all—acute bacterial infections cause increased activity, and that this activity can be induced in several different mutants, suggests that the effect of infection on locomotor behaviour is a common phenomenon and may represent a complex trait emerging as the result of the induction of multiple molecular pathways.

### Increased activity is not a moribund behaviour and is affected by immune activation

We became particularly interested in the increased activity observed during infection with *M. luteus* because unlike the other bacteria examined, increased activity following injection with *M. luteus* was transient, progressively diminishing over the three days following infection,

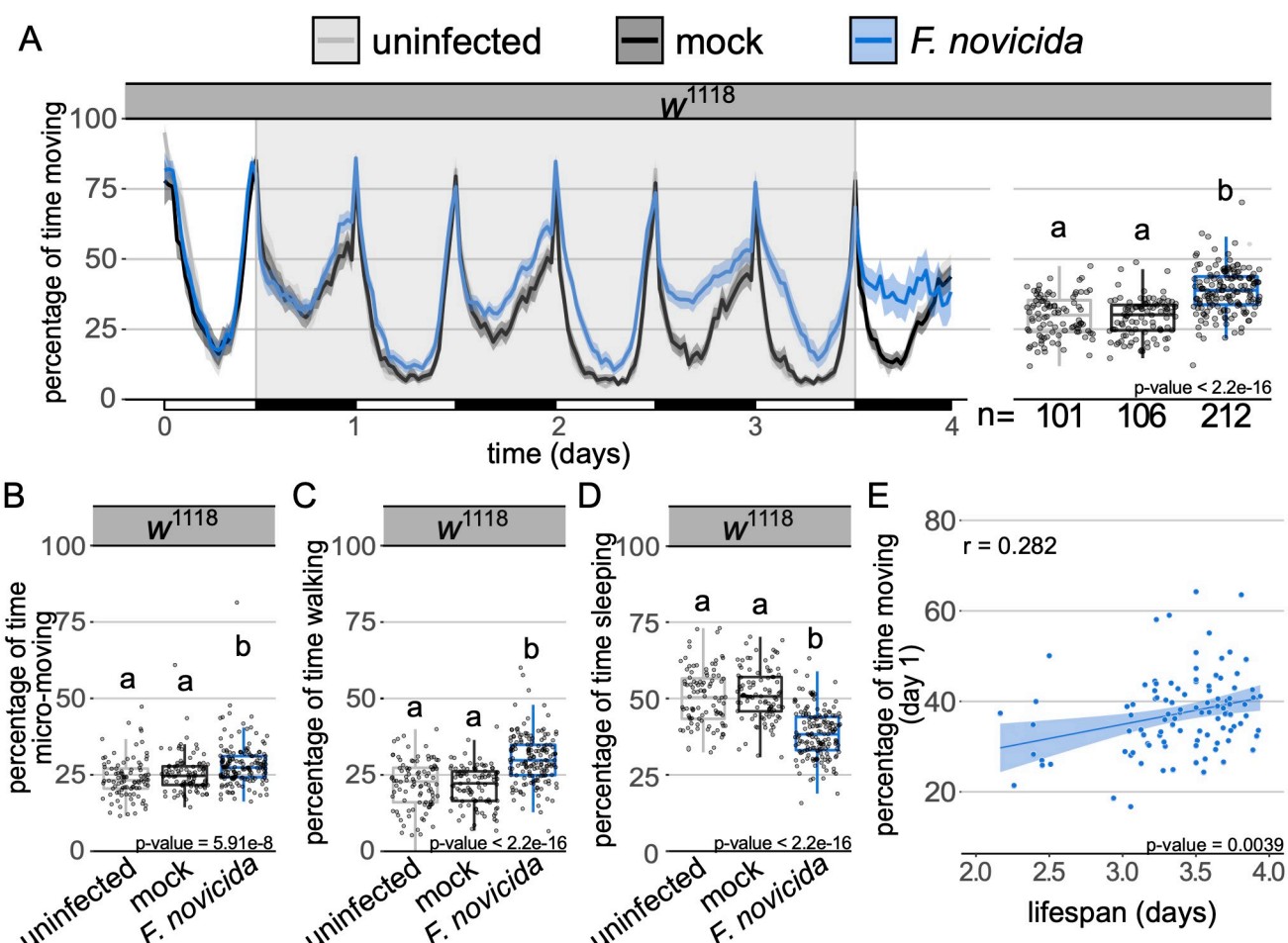

**Fig 1. Infection with *Francisella novicida* leads to increased locomotor activity. (A)** Ethogram showing percentage of time wild-type males spend moving over time in 30-min bins. Alternating white and black horizontal bar along the x-axis indicates day (12h light) and night (12h dark) cycles, respectively. Uninfected and mock controls are represented by grey and black tracings, respectively. Infected flies are in blue. Shaded areas surrounding solid lines represent the 95% confidence intervals. Flies were injected within two hours of the beginning of their light cycle (t = 0). Background area highlighted in grey indicates the time for which data were analysed as represented in adjoining boxplot. Boxplots showing the percentage of time wild-type males **(B)** engage in micromovements (e.g. feeding and grooming), **(C)** walking and **(D)** sleeping. Markers indicate individual data points. Horizontal bar within each box represents the median. The bottom and top lines of the box represent the 1$^{st}$ and 3$^{rd}$ quartiles, respectively. Whiskers represent the smallest value between: highest and lowest values or 1.5x the interquartile range. Boxes without common letters are significantly different. Sample sizes (n) are indicated under the boxplots. Plots throughout have identical formatting, therefore a full description of ethogram and boxplot features is omitted in subsequent legends. *Francisella novicida* infected animals **moved significantly more** than both the uninfected and mock controls (Kruskal-Wallis chi-square = 99.206, df = 2, n = 419, p = 2.2e-16; Dunn's *post hoc*: mock|*F. novicida* = 1.2e-17, mock|uninfected = 0.41, uninfected|*F. novicida* = 1.4e-14). Infected flies **engaged in more micromovements** (Kruskal-Wallis chi-square = 33.287, df = 2, n = 419, p = 5.9e-08; Dunn's *post hoc*: mock|*F. novicida* = 9.5e-05, **walked more** (Kruskal-Wallis chi-square = 88.383, df = 2, n = 419, p = 2.2e-16; Dunn's *post hoc*: mock|*F. novicida* = 1.02e-15, mock |uninfected = 0.48, uninfected|*F. novicida* = 2.7e-13), mock |uninfected = 0.23, uninfected|*F. novicida* = 2.7e-07), and **spent less time sleeping** (Kruskal-Wallis chi-square = 99.206, df = 2, n = 419, p = 2.2e-16; Dunn's *post hoc*: mock|*F. novicida* = 1.2e-17, mock|uninfected = 0.41, uninfected|*F. novicida* = 1.4e-14) than both the mock and uninfected controls. **(E)** Plot depicts the correlation between the percentage of time spent active on the first day of *F. novicida* infection and the lifespan of each fly that survived between 2 and 4 days (Pearson's correlation, *r* = 0.282; t = 2.96, df = 101, p = 3.9e-03). Data from multiple replicates are shown.

until activity levels returned to baseline (Fig 2A). That flies spend more time active during *M. luteus* infection is particularly important because it demonstrates that infection-induced activity is not a moribund behaviour: *M. luteus* infection is cleared by the immune response and flies are not killed by this infection over the following four days; this contrasts with *F. novicida*,

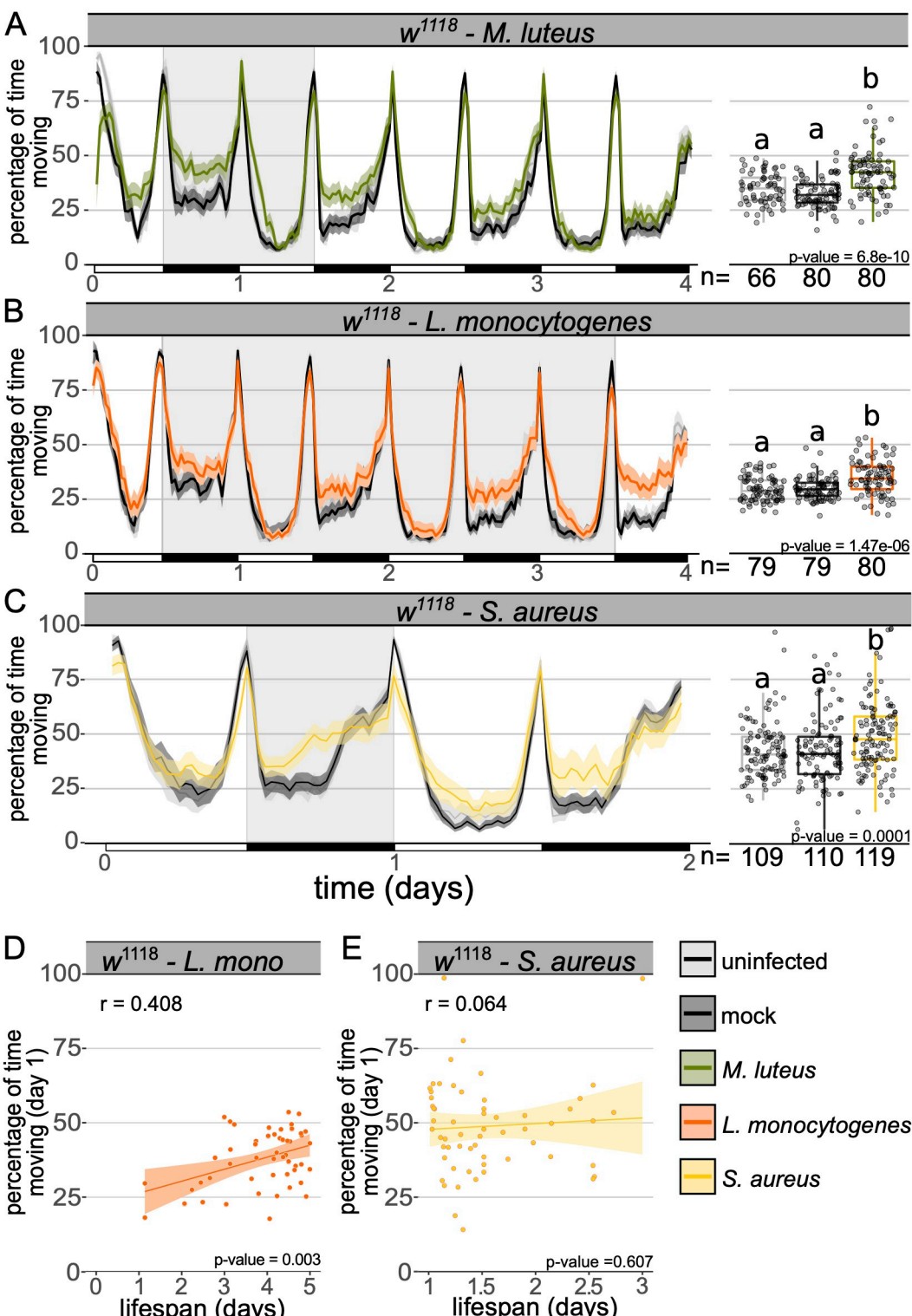

**Fig 2. Infection with multiple bacteria leads to increased activity in wild-type flies.** Ethogram showing percentage of time wild-type flies spend moving over time in 30-min bins during infection with (**A**) *Micrococcus luteus* (**B**) *Listeria monocytogenes* and (**C**) *Staphylococcus aureus*. Infected flies moved significantly more than both the uninfected and mock controls (***M. luteus***: Kruskal-Wallis chi-square = 42.22, df = 2, n = 226, p = 6.8e-10; Dunn's *post hoc*: mock|*M. luteus* = 7.02e-10, mock|uninfected = 0.07, uninfected|*M. luteus* = 3.0e-05; **L. monocytogenes**: Kruskal-Wallis chi-square = 26.859, df = 2, n = 238, p = 1.5e-06; Dunn's *post hoc*: mock|*L.monocytogenes* = 3.05e-05, mock|uninfected = 0.68, uninfected|*L*.

*monocytogenes* = 8.7e-06; ***S. aureus***: Kruskal-Wallis chi-square = 14.006, df = 2, n = 338, p = 0.001; Dunn's *post hoc*: mock|*S. aureus* = 3.10e-03, mock|uninfected = 0.91, uninfected|*S. aureus* = 2.43e-03). (D-E) Plots depict the correlation between lifespan and the percentage of time spent active for each fly during the first day of infection with **(D)** *L. monocytogenes* or **(E)** *S. aureus* (*L. monocytogenes*: Pearson's correlation, *r* = 0.408; t = 3.13, df = 49, p = 2.9e-03; *S. aureus*: Pearson's correlation, *r* = 0.064; t = 0.516, df = 65, p = 0.607). Behavioural assays were performed at least twice, data from all replicates are shown.

*L. monocytogenes* and *S. aureus*, all of which kill more than half of all infected flies within four days (S6A–S6J Fig).

The observation that the rapidly cleared pathogen, *M. luteus*, induced increased activity, prompted us to test whether bacterial detection by immune pathways and the subsequent signalling was driving this behaviour. Previous work found that the NFκB transcription factor RELISH which plays a vital role in *D. melanogaster's* immune response, is required for infection-induced sleep [27]. We infected flies lacking the immune deficiency (IMD) and Toll pathways, the primary microbe-detection pathways in *D. melanogaster* [38,40,46,50]. We found that ablation of IMD (*imd*[10191]) and Toll signalling (*spz*[eGFP]) had disparate effects on locomotion during infection. Activity during *M. luteus* infection was unaffected in *imd* mutants, but no increase in activity was observed following *M. luteus* infection in *spz* mutants (Fig 3A and 3B), in keeping with the fact that *M. luteus* is primarily an agonist of the Toll pathway [37,51,52]. To confirm this finding, we repeated this experiment using flies carrying a different *spz* allele [53] (Fig 3C) and with an additional Gram-positive pathogen, *S. aureus* (S7 Fig), both of which confirmed our observation that Toll signalling is required for increased activity during infection with these Gram-positive bacteria.

The absence of increased activity in *M. luteus* infected Toll mutants indicates that Toll signalling is required for increased activity during this infection. However, mutation of either *imd* or *spz*, as well as the combination of the two (*imd*[10191]; *spz*[eGFP]), did not affect the increase in activity caused by *F. novicida* (S8A–S8C Fig). These findings demonstrate that in *F. novicida* infections, the activity phenotype is independent of *Toll* and IMD pathways. The IMD pathway is thought to be mostly responsive to Gram-negative bacteria, but previous work has shown that it is involved in the recognition of *L. monocytogenes* [54]. We found that *L. monocytogenes* infection led to increased activity in *imd* mutants, further supporting the IMD independence of this response (S8D Fig). The dependence on Toll for increased activity in *M. luteus* but not *F. novicida* infections, demonstrates that infection induces activity via different signalling pathways during these infections.

The independence from IMD signalling of infection-induced activity was surprising given that in both *M. luteus* and *F. novicida*-infected flies, increased activity was roughly correlated with bacterial load (S6K and S6L Fig), suggesting that this phenotype is responsive to bacterial recognition systems. Flies infected with *M. luteus* showed increased activity during the early stages of infection when bacterial numbers were high, and a return to baseline levels of activity following bacterial clearance. Similarly, in *F. novicida*-infected flies, activity increased in parallel with bacterial load. We infected flies with heat-killed bacteria to see if bacterial detection alone could induce activity. We found that while heat-killed *M. luteus* induced activity, heat-killed *F. novicida* did not (S9A and S9B Fig), further supporting the idea that activity is regulated via different pathways during these infections, and that Toll signalling plays a vital role during infection with *M. luteus*.

To better understand the role of *F. novicida* detection on increased activity, we orally administered the antibiotic tetracycline, which results in a persistent, low number of *F. novicida* over several days [44], to *F. novicida*-infected flies. Administration of tetracycline eliminated infection-induced activity (S10A and S10B Fig). Importantly, tetracycline treatment

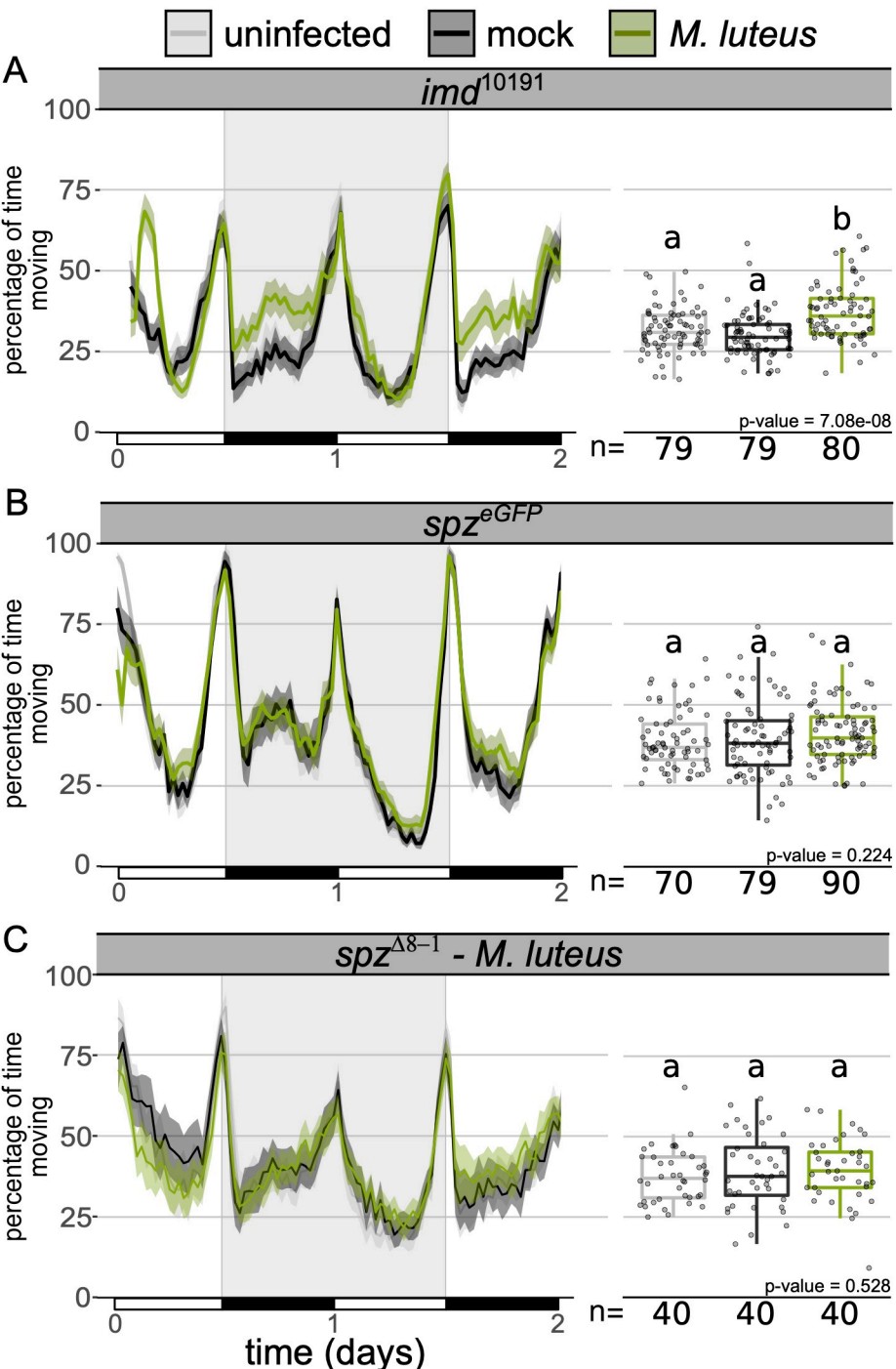

**Fig 3. Toll signalling mutants do not increase activity during *M. luteus* infection.** Ethogram showing percentage of time (**A**) *imd*[10191], (**B**) *spz*[eGFP] and (**C**) *spz*[Δ8−1] flies spend moving over time in 30-min bins. *Micrococcus luteus* infected *imd*[10191] flies–but not *spz*[eGFP]–moved significantly more than both the uninfected and mock controls (*imd*[10191]: Kruskal-Wallis chi-square = 32.93, df = 2, n = 238, p = 7.1e-08; Dunn's *post hoc*: mock|*M. luteus* = 6.9e-08, mock|uninfected = 0.1, uninfected| *M. luteus* = 1.07e-04; ***spz***[eGFP]: Kruskal-Wallis chi-square = 2.99, df = 2, n = 239, p = 0.22; ***spz***[Δ8−1]: Kruskal-Wallis chi-square = 1.28, df = 2, n = 120, p = 0.528). Data from multiple replicates are shown.

does not kill bacteria, but rather inhibits bacterial growth [55], thus providing an interesting compliment to the heat-killed assay in which we also saw the ablation of infection-induced activity. Together these findings suggests that live, proliferating bacteria, play a vital role in triggering activity during *F. novicida* infection. As predicted, when infected with a tetracycline resistant strain of *F. novicida* (tetR), tetracycline treated flies once again exhibited increased levels of activity (S10C Fig). Reducing the initial inoculum by a tenth (~170CFUs) resulted in a near-identical activity profile–and importantly, timing of activity onset–as the higher dose (S9C Fig), suggesting that live, proliferating bacteria may be more important than absolute numbers in inducing activity.

## Infection causes time-specific metabolic dysregulation, but increased activity is not a response to starvation

Infection with *M. luteus* inhibits insulin signalling, as evidenced through a reduction in phosphorylated AKT, and this metabolic shift is concomitant to a reduction in triglyceride levels [56]. Similarly, infection with *F. novicida* leads to triglyceride loss as well as hyperglycaemia and reduced levels of glycogen [44]. The interplay between immune and metabolic signalling pathways is thought to be indicative of the metabolic burden associated with infection and the need to redistribute available resources [56–58]. We therefore surmised that the metabolic shifts observed during *M. luteus* and *F. novicida* infection could play a role in infection-induced activity, despite the difference in the requirement of Toll signalling, and sought to determine whether *F. novicida* and *M. luteus* infections led to similar metabolic phenotypes.

We tested whether infection with *F. novicida* inhibits insulin signalling as has been previously reported for *M. luteus* infection [56]. We found that infection with *F. novicida* inhibits insulin signalling as determined through the observance of lower levels of phosphorylated-AKT during late infection (72-80h post-injection; Fig 4A), these flies were also hyperglycaemic and exhibited depleted triglyceride and glycogen stores (Fig 4B). Similarly, late in infection with *M. luteus*, flies had lower triglyceride and glycogen levels, but no change in circulating sugars. During early infection (24-30h post-injection), we observed hypoglycaemia with *F. novicida*, but not *M. luteus*, a significant reduction of triglycerides with *M. luteus*, but not *F. novicida*, and low levels of glycogen in both infections (Fig 4B). These results confirm previous work showing that bacterial infection can lead to metabolic pathology, including hyperglycaemia and loss of triglyceride and glycogen stores, and that these effects are often limited to specific times over the course of infection [44,57,59].

Because infected flies exhibit starvation-like effects on metabolite stores and insulin-pathway activity, we tested whether hyperactivity during infection was linked to infection-induced starvation signalling. Infection-induced anorexia has been observed in both mammals and insects [16,19,60,61] and hyperactivity is a known consequence of starvation in *D. melanogaster* [62–65]. We tested whether our two focal infections–*F. novicida* and *M. luteus*–also resulted in reduced food consumption. Surprisingly, we found that *F. novicida* infected flies consumed 19.4% more food during infection compared to their mock controls and that food consumption was unaffected by infection with *M. luteus* (Fig 4C). Thus, infection-induced activity does not appear to be a by-product of anorexia.

Next, we tested the possibility that infection-induced increases in activity could be a product of endocrine signalling disruptions mimicking the effects of starvation. In *D. melanogaster*, starvation increases activity via *adipokinetic hormone* (AKH) signalling in neurons [63,65]. We infected flies with a pan-neural reduction of the adipokinetic hormone receptor (*nSyb>AkhR*-IR), a strategy previously shown, and confirmed here, to obliterate starvation-induced hyperactivity (S11 Fig) [65]. Neuronal knockdown of *AkhR* did not affect activity during either of *F.*

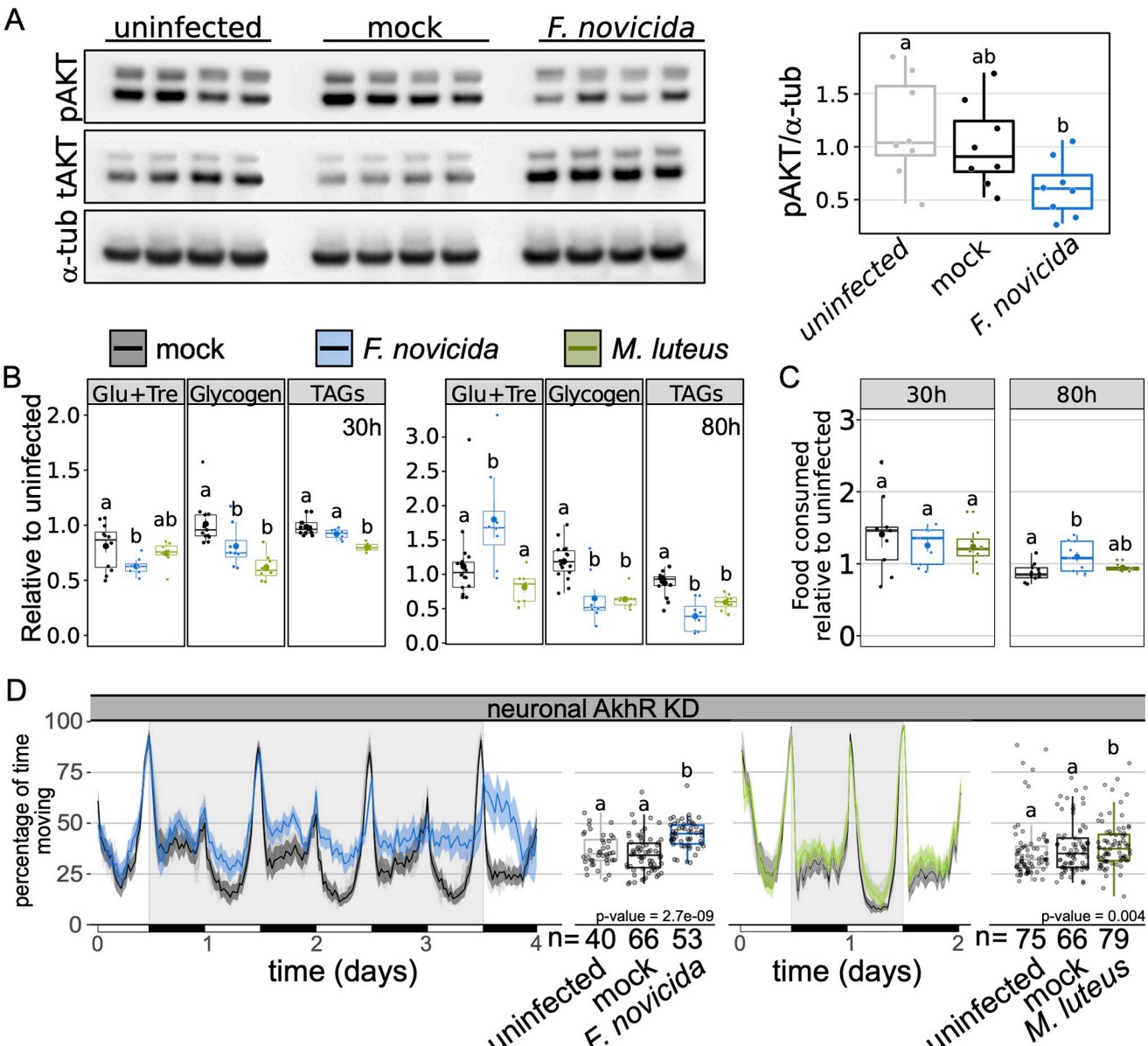

**Fig 4. *Micrococcus luteus* and *Francisella novicida* infection similarly disrupt host metabolism. (A)** Western blot of phosphorylated AKT (Ser505) during *F. novicida* infection in wild-type flies. Total AKT and tubulin levels for each sample also shown. Boxplot shows quantification of pAKT relative to α-tubulin using data from both repeats of the experiment. Kruskal-Wallis chi-square = 6.665, df = 2, n = 24, p = 0.036; Dunn's *post hoc*: mock|*F novicida* = 0.091, mock|uninfected = 0.548, uninfected| *F. novicida* = 0.04;) **(B)** Levels of circulating and stored glycogen and triglyceride (TAG) and feeding activity **(C)** during early (30h) and late (80h) infection. Mock controls are indicated in grey, whilst *F. novicida* and *M. luteus* injections are indicated in blue and green, respectively. Data are plotted relative to the mean of uninfected controls. **Metabolism 30h post infection**: There was a significant effect of infection, such that glucose/trehalose levels were significantly lower in *F. novicida*-infected flies (AOV: df = 2, n = 28, F = 3.449, p = 0.048; Tukey's HSD: mock|*F. novicida* = 0.038, mock |*M. luteus* = 0.62, *M. luteus*|*F. novicida* = 0.29). Infection had a significant effect on glycogen stores which were reduced in both infections (AOV: df = 2, n = 29, F = 12.112, p = 2.1e-04; Tukey's HSD: mock|*F. novicida* = 0.049, mock |*M. luteus* = 1.4e-04, *M. luteus*|*F. novicida* = 0.102), but only *M. luteus*-infected flies had a significant reduction in triglycerides (AOV: df = 2, n = 27, F = 18.763, p = 1.5e-05; Tukey's HSD: mock|*F. novicida* = 0.073, mock|*M. luteus* = 8.8e-06, *M. luteus*|*F. novicida* = 3.5e-03). **Metabolism 80h post infection**: There was a significant effect of infection, such that glucose levels were significantly higher in *F. novicida*-infected flies (AOV: df = 2, n = 31, F = 6.883, p = 3.5e-03; Tukey's HSD: mock|*F. novicida* = 0.018, mock|*M. luteus* = 0.41, *M. luteus*|*F. novicida* = 3.4e-03). Infection led to a significant reduction of both glycogen (AOV: df = 2, n = 31, F = 17.315, p = 1e-05; Tukey's HSD: mock|*F. novicida* = 1.6e-04, mock |*M. luteus* = 1.1e-04, *M. luteus*|*F. novicida* = 0.99), and triglycerides (AOV: df = 2, n = 28, F = 21.622, p = 2.5e-06; Tukey's HSD: mock|*F. novicida* = 2.3e-06, mock|*M. luteus* = 2.7e-07, *M. luteus*|*F. novicida* = 0.066). **Feeding**: Neither infection affected feeding within **30h** of injection (AOV: df = 2, n = 26, F = 1.117, p = 0.35), but **80h** post-injection, *F. novicida*-infected flies fed significantly more than mock controls but not more than *M. luteus*-infected flies (AOV: df = 2, n = 29, F = 7.289, p = 4.2e-03; Tukey's HSD: mock|*F. novicida* = 9.2e-03, mock|*M. luteus* = 0.58, *M. luteus*|*F. novicida* = 0.082). **(D)** Ethogram showing percentage of time flies spend moving over time in

30-min bins. Neuronal KD of adipokinetic hormone did not eliminate *F. novicida* or *M. luteus* infection-induced activity (**F. novicida**: Kruskal-Wallis chi-square = 39.461, df = 2, n = 159, p = 2.7e-09; Dunn's *post hoc*: mock|*F. novicida* = 4.3e-09, mock|uninfected = 0.35, uninfected|*F. novicida* = 1.4e-05; **M. luteus**: Kruskal-Wallis chi-square = 10.825, df = 2, n = 233, p = 0.004; Dunn's *post hoc*: mock|*M. luteus* = 0.009, mock|uninfected = 0.817, uninfected| *M. luteus* = 0.009). Data from multiple replicates shown.

*novicida* or *M. luteus* infection, confirming that the infection-induced increase in activity observed during these infections is distinct from a starvation response (Fig 4D). This finding is important because one of the proposed advantages of hyperactivity during starvation is greater resource acquisition from increased foraging. Thus, despite the failure of this infection to induce a starvation-like response via AKH, the results of these experiments are consistent with the idea that increased activity during *F. novicida* infection could lead to greater resource acquisition through increased feeding.

## Fat body derived *spz* contributes to *M. luteus* infection-induced activity

Bacterial peptidoglycan activates the Toll and IMD pathways, leading to the synthesis and secretion of antimicrobial peptides (AMPs) by the fat body in *D. melanogaster* [36,38,46,50]. The production of AMPs contributes to the control of most bacterial infections. *spz* is synthesized and secreted as an inactive pro-protein where extracellular recognition factors initiate protease cascades which result in its proteolysis to produce active *spz*. This ligand binds and activates cell-surface Toll receptors [66–69]. Since the Toll pathway is activated by Gram-positive bacteria [40,70], and we found that mutants of this pathway do not show an increase in activity during infection with the Gram-positive bacteria *M. luteus* and *S. aureus* (Fig 3B and 3C, and S7 Fig), we predicted that Toll signalling in the fat body played a role in infection-induced activity. To test this, we infected flies carrying fat body knockdowns of *spz* (*c564>spz*-IR), the circulating ligand that directly activates Toll; *MyD88* (*c564>MyD88*-IR), a key adaptor in the Toll pathway; and *Dif* (*c564>Dif*-IR), the primary Toll-activated NF-KB transcription factor in adult *Drosophila* [36,46,69]. As observed in the whole-body Toll signalling mutants, restricted knockdown of the Toll ligand *spz* to the fat body completely abolished the increase in activity observed during *M. luteus* infection (Fig 5A). Flies with *MyD88* or *Dif* knocked down in the fat body were not significantly more active than uninfected controls during *M. luteus* infectio, though they did show elevated activity relative to mock-infected controls (Fig 5B–5D). These findings demonstrate that fat body-derived *spz* and fat body Toll pathway activity play a crucial role in the modulation of locomotor activity during infection.

## Neuronal KD of Toll signalling does not affect activity during *M. luteus* infection

Given that neither *spz* mutants nor *spz* fat body KD flies exhibit infection-induced activity, we thought that fat body-derived *spz* could be acting on other tissues to induce this behaviour and decided to test whether neuronal knockdown of Toll signalling would also affect the activity phenotype. We infected flies with a pan-neural reduction of either *MyD88* (*nSyb> MyD88*-IR) or *Dif* (*nSyb>Dif*-IR) and found that these flies still exhibited increased activity in *M. luteus* infection (Fig 5E and 5F). This could reflect other mechanisms acting in parallel or it could be due to residual *MyD88/Dif* function in neurons. Additionally, the genetic controls (*nSyb>+*) exhibited surprisingly high levels of activity, with nearly 50% of all flies moving, regardless of treatment. These flies also failed to exhibit a consistent phenotype; *M. luteus*-infected animals were significantly more active than uninfected, but not more than mock-injected, controls (Fig 5G). The inconsistent effect of infection on control flies, and the possibility of incomplete

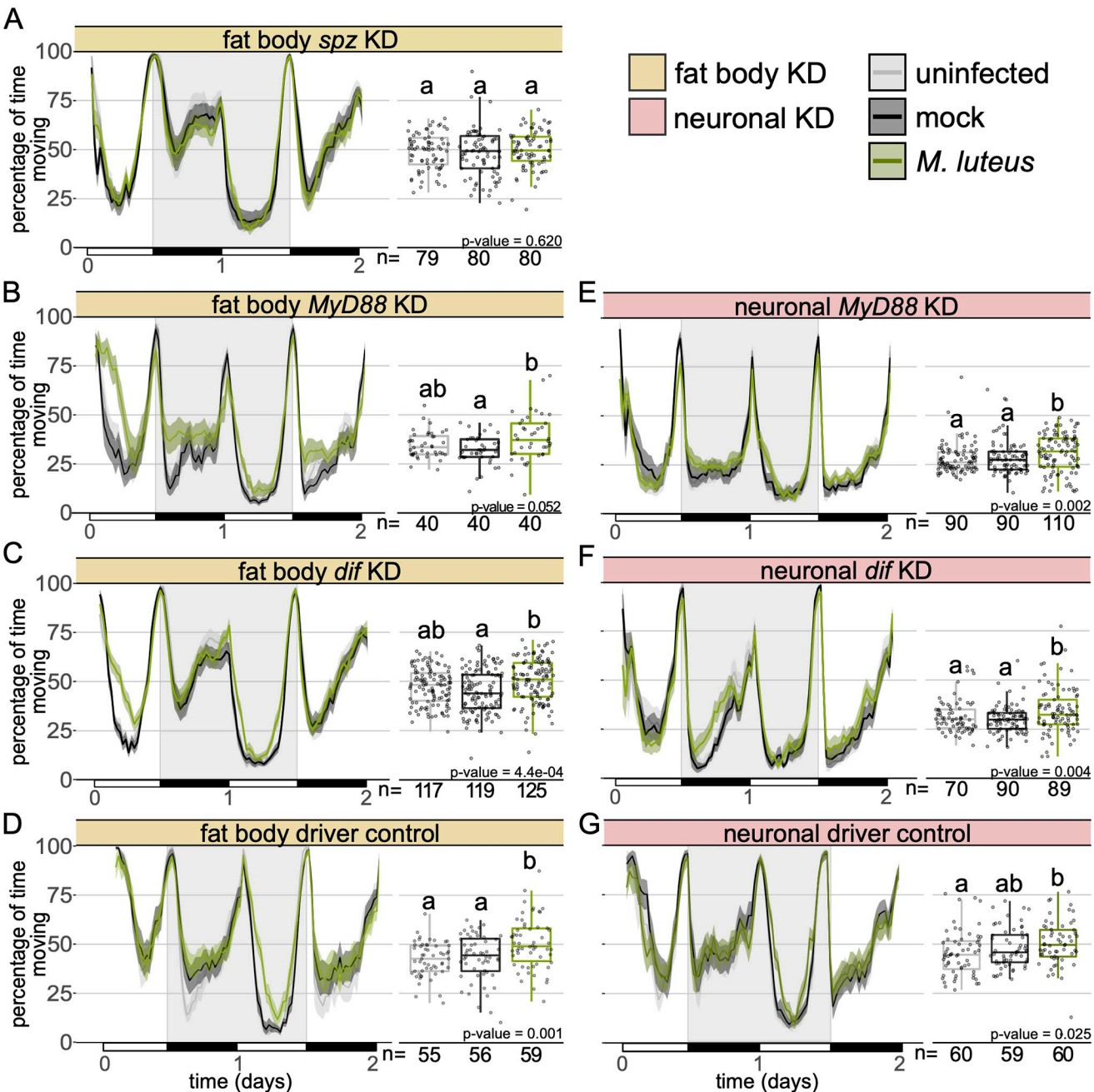

**Fig 5. Fat body Toll signalling is required for infection-induced activity during *Micrococcus luteus* infection.** Ethogram showing percentage of time flies spend moving over time in 30-min bins with fat body **(A)** *spz* (w; *c564>* w; spz-IR), **(B)** *MyD88* (w; *c564>* w; *MyD88*-IR) and **(C)** *Dif* (w; *c564>* w; *Dif*-IR) knockdown or **(D)** driver control (w; *c564>*+). *Micrococcus luteus* infection had no effect on *spz*, nor *MyD88* fat body KD flies (***c564>spz*-IR**: Kruskal-Wallis chi-square = 0.96, df = 2, n = 239, p = 0.62; ***c564>MyD88*-IR**: Kruskal-Wallis chi-square = 5.92, df = 2, n = 120, p = 0.052). *Dif* fat body KD flies and the genetic control flies infected with *M. luteus* were significantly more active than mock-injected but not uninfected controls (***c564>Dif*-IR**: Kruskal-Wallis chi-square = 15.45, df = 2, n = 361, p = 4.4e-04; Dunn's *post hoc*: mock|*M. luteus* = 2.6e-04, mock|uninfected = 0.077, uninfected| *M. luteus* = 0.053; ***c564>*+**: Kruskal-Wallis chi-square = 13.485, df = 2, n = 170, p = 0.001; Dunn's *post hoc*: mock|*M. luteus* = 0.017, mock|uninfected = 0.303, uninfected|*M. luteus* = 0.001). Pan-neural **(E)** *MyD88* (*nSyb>MyD88*-IR) and **(F)** *Dif* knockdown (*nSyb>Dif*-IR) led to increased activity during infection (***nSyb>MyD88*-IR**: Kruskal-Wallis chi-square = 12.69, df = 2, n = 290, p = 1.8e-03; Dunn's *post hoc*: mock|*M. luteus* = 0.027, mock|uninfected = 0.304, uninfected| *M. luteus* = 1.7e-03; ***nSyb>Dif*-IR**: Kruskal-Wallis chi-square = 11.05, df = 2, n = 249, p = 3.9e-03; Dunn's *post hoc*: mock|*M. luteus* = 5.7e-03, mock|uninfected = 0.68, uninfected| *M. luteus* = 0.018). **(G)** Infected neuronal driver controls moved significantly more than uninfected controls but no more than mock controls (nSyb>+: Kruskal-Wallis chi-square = 7.35, df = 2, n = 179, p = 0.025; Dunn's post hoc: mock|M. luteus = 0.19, mock| uninfected = 0.24, uninfected|M. luteus = 0.021). Data from multiple replicates are shown.

knockdown, preclude any assertion regarding the role of neuronal Toll signalling in infection-induced changes in activity.

## Discussion

Here we show that several bacterial infections lead to a marked increase in activity in *D. melanogaster*. This enhanced level of activity is mostly explained by an increase in walking (Fig 1). Though several bacteria induce activity upon infection in multiple fly lines with mutations in their immune response, we see pathogen/immune pathway specificity, as mutations in Toll signalling ablate activity-induction by some, but not all, bacteria. Finally, we demonstrate that fat body-derived *spz* is required for this activation.

Immune activation has been shown to affect a range of behaviours and physiological functions including sleep, reproduction, cognition and metabolism [27,28,33,57,59,71–73]. Infection-induced changes in the host are often thought to be of benefit to either the host or the pathogen. Pathogen-mediated changes in host behaviour can lead to decreased survival, transmission and terminal host localisation [9,11,13,74], whilst host-mediated changes during infection have been found to result in improved resistance and colony/conspecific protection [75–78]. We found that activity level on the first day of infection had a strong correlation with survival, though this relationship was not observed across all bacterial strains (Figs 1E and 2D and 2E). Increased activity at the onset of infection could lead to increased survival if hyperactive animals are likely to locate preferred habitats [79], or food resources to counter the effects of infection [80].

One well-documented change in host behaviour resulting from infection is anorexia [16,19,60]. Animals often decrease feeding in response to infection and this behaviour can be either harmful or beneficial to the host. In addition to infection-induced anorexia, *D. melanogaster* exhibits striking wasting phenotypes in a number of infections, characterized by decreased levels of glycogen and triglycerides [44,57,59,81]. Given the strong associations between infection and resource acquisition and utilization, one could imagine a scenario in which rather than infection directly leading to increased activity, instead, the metabolic dysregulation caused by infection signalled for increased activity as a means to acquire more nutrition [64,65]. Whilst both *M. luteus* and *F. novicida* infections led to strong wasting phenotypes, we found no evidence that starvation signalling contributed to increased activity, though increased activity was associated with greater food intake during *F. novicida* infection (Fig 4C).

While previous work found that bacteria-infected flies have poor quality sleep, as assessed through number of sleep bouts and bout duration [28], these flies were not found to be more active than healthy controls. Infection with Gram-negative bacteria in *D. melanogaster* yields contrasting findings, showing that these infections can both reduce and increase sleep [17,27,28]. One study found that increased sleep led to greater survival and bacterial clearance, but the flies in this study were sleep deprived prior to infection, making it difficult to disambiguate the effect of the earlier sleep deprivation from–and on–subsequent infection and compensatory sleep. Interestingly, in that same study, flies in the control treatment–which were not sleep deprived–appeared to exhibit increased activity following infection [17]. We attribute the discrepancy between the current study and previous work in observing a change in activity, to differences between the annotative capabilities of the different activity monitoring systems used, specifically, the greater spatial and temporal resolution afforded by the method employed here [41]. Another potential explanation for the discrepancy is that while previous work evaluated activity levels immediately after infection [17,27], we focused our attention on activity

levels several hours or even days following initiation of systemic infection. Thus, the temporal dynamics of the infection and its impact on behaviour may be intimately related.

The duration of increased activity differed markedly between our two primary pathogens, *M. luteus* and *F. novicida* (Figs 1 and 2). Not only did flies infected with *M. luteus* appear to exhibit earlier onset of activity than *F. novicida* -infected flies, but they also returned to 'normal' levels of activity 3 days post-infection (Fig 2). In contrast, flies infected with *F. novicida* exhibit a constant, if not amplifying, increase in activity as infection progresses (Fig 1). Even with a tenth of the *F. novicida* inoculum, we observed near-identical timings and patterns with respect to the onset and progression of activity (S9C Fig). Furthermore, when bacterial numbers were kept relatively low due to the presence of the antibiotic, tetracycline, no increase in activity was observed, suggesting that in this infection, activity is induced by either some amount of bacteria exceeding 3000 CFUs [44], or a by-product of active bacterial proliferation. Correspondingly, heat-killed *F. novicida* also failed to induce activity whilst heat-killed *M. luteus* did (S9 Fig). The finding that heat-killed *M. luteus* leads to increased activity lends further support to our hypothesis that in this infection, Toll pathway signalling is involved in activity induction.

Given the role of the fat body in the immune response, we predicted that pathogen recognition and subsequent activation of immune signalling pathways could contribute to the observed increase in activity. In contrast to Toll, IMD signalling was not necessary for increased activity (Fig 3 and S8 Fig). This IMD-independence could explain the lack of effect that we observed during *E. coli* and *E. cloacae* infection (S4 Fig). Infection with both *F. novicida* and *M. luteus* led to decreased glycogen and triglyceride levels (Fig 4). Previous work found that infection with *E. coli* leads to triglyceride loss, but in contrast to what we observed here, *E. coli* infection did not affect glycogen stores, nor circulating sugars–as observed during *F. novicida* infection [81]. Thus, metabolic dysregulation, specifically, disruptions to insulin signalling as has been shown in both *F. novicida* and *M. luteus* infections [56], could play a role in infection-induced activity.

Intercellular signalling via cytokines has been shown to be vital to the induction of sickness behaviours [1,82–84]. Thus, immune detection in any of an organism's organs has the potential to send signals to the brain that ultimately affect behaviour. Fat body-derived *spz* has been shown to be sufficient to induce sleep following infection, in a different experimental paradigm than the one we use here [27]. Furthermore, a recent study showed that knocking down Toll signalling in the sleep-regulating R5 neurons suppressed the characteristic increase in sleep that is observed following sleep deprivation [85]. Similarly, the antimicrobial peptide *nemuri* has been shown to be an infection-inducible factor able to promote sleep [86]. Collectively these findings suggest a model wherein *spz* originating from the fat body acts on a group of heretofore unidentified neurons to induce behavioural changes during infection. However, we are unable to observe a clear requirement for neuronal *MyD88* or *Dif* in triggering increased activity after infection; it is unclear whether this is caused by inadequate knockdown in a relevant neuronal population, no requirement for these factors in the responding cell, or the responding cells not being neurons. What is clearer is that, whilst *spz* is secreted from tissues other than the fat body, the contribution of fat body derived *spz* is necessary for the activity phenotype. Our results leave open the question of which tissues are relevant targets of this signalling in inducing this behavioural change.

The role of increased activity during infection remains elusive but given that the behaviour appears to be activated via multiple pathways suggests that it serves a function rather than being the result of pleiotropy. That females did not exhibit increased activity during infection is intriguing and worthy of further exploration. The sexes have been found to differ dramatically in the timing, magnitude, behaviour, and nature of their response to infection

[32,73,81,87–93], and these differences are likely to play a role in the discrepancy here observed. Future work to discover other mechanisms involved has the potential to address the question of underlying function. Furthermore, the diversity of bacteria as well as the tools available to manipulate bacterial genomes, can be used to identify bacteria-derived signals that contribute to this response.

## Methods

### General experimental procedures

$w^{1118}$ flies were used as wild-type flies throughout the study. A complete record of all other fly lines used in this study can be found in the supplementary information (S3 Table). For all experiments, male flies were collected following eclosion and kept in same-sex vials for 5–7 days in groups of 20. Thus, all experiments were conducted on flies between 5 and 8 days old. Flies were maintained on a standard diet composed of 10% w/v yeast, 8% w/v fructose, 2% w/v polenta, 0.8% w/v agar, supplemented with 0.075% w/v nipagin and 0.0825% vol propionic acid, at 25˚C. Bacteria were grown from single colonies overnight at 37˚C shaking with the exception of *L. monocytogenes* which was grown at 37˚C without shaking. Each fly was injected in the lateral anterior abdomen with 50 nanolitres of bacteria diluted in PBS (OD = 1 for *E coli*, *E cloacae*, and *M luteus*; OD = 0.1 for *F. novicida* and *L. monocytogenes*; OD = 0.001 for *S. aureus*). Control flies were either injected with sterile PBS or were anaesthetized but otherwise unmanipulated, here referred to as mock controls and uninfected, respectively. Heat-killed bacteria were prepared as described above and then incubated for 10 minutes at 90˚C. Injections were carried out using a pulled-glass capillary needle and a Picospritzer injector system (Parker, New Hampshire, US). Following injection flies were kept at 29˚C on our standard diet. Flies given tetracycline were transferred to 0.04% tetracycline food (standard diet, supplemented with powdered Tetracycline≥98.0% (NT), 87128 Sigma-Aldrich). Tetracycline-resistant (tetR) *F. novicida* was a gift from Thomas Henry's laboratory in Lyon.

### Behavioural experiments

For all experiments, flies were sorted into glass tubes [70 mm × 5 mm × 3 mm (length × external diameter × internal diameter)] containing the same food used for rearing. After 2 days of acclimation under a regime of 12:12 Light:Dark (LD) condition in incubators set at 25˚C animals were subject to either bacterial injection, mock injection, or anaesthetization (as above described), between zeitgeber time (ZT) 00 to ZT02 (the first 2h after lights ON) and transferred to fresh glass tubes containing our lab's standard food as described above. Activity recordings were performed using ethoscopes [41] under 12:12 LD condition, 60% humidity at 29˚C. Behavioural data analysis was performed in RStudio [94] employing the Rethomics suite of packages; Rethomics permits automated analysis of time spent undertaking large-scale movements (walking) as well as micro-movements (grooming, feeding, spinning, etc.) [41]. All behavioural assays were repeated at least twice with 20–60 flies/treatment/experiment. For lethal infections, behavioural data were analysed for the period between the first and final 12h of the assay; these windows of time were excluded as they encompass excessive noise due to awakening from anaesthesia/acclimation and mortality leading to declining sample sizes, respectively. For non-lethal infections, we analysed the 24h period following the initial 12h (t = 12h – 36h post infection), this time encompasses the duration of *M. luteus* infection after which live bacteria are no longer detected in flies.

## Bacterial quantification

Bacteria were quantified either via qPCR (*F. novicida*) or plating (*M. luteus*). For plating, one fly was homogenised in 100μl of sterile ddH$_2$O. Homogenates were serially diluted and plated onto LB agar plates where they incubated for 16-18h. Following incubation, the number of individual bacterial colonies observed on each plate was quantified and calculated to determine the number of CFUs present in each fly. For qPCR, one fly was homogenised in a 100μl of Tris-EDTA, 1% Proteinase K (NEB, P8107S) solution. Homogenates were incubated for 3h at 55˚C followed by a ten-minute incubation at 95˚C. Following incubation, we performed our qPCR protocol as described elsewhere [44] to determine the number of bacterial colony forming units (CFU). All quantifications were repeated at least twice with 8–16 samples/treatment/experiment.

## Measurement of triglycerides

Triglycerides were measured using thin layer chromatography (TLC) assays as described elsewhere [95]. Briefly, each sample consisted of 10 flies; flies were placed in microcentrifuge tubes and stored at -80˚C until the time of analysis. To perform the TLC assay, samples were removed from the -80˚C freezer and spun down (3 min at 13,000 rpm at 4˚C) in 100μl of a chloroform (3): methanol (1) solution. Flies were then homogenised and subject to a further 'quick spin'. Standards were generated using lard dissolved in the same chloroform: methanol solution. We loaded 2μl of each standard and 20μl of each sample onto a silica gel glass plate (Millipore). Plates were then placed into a chamber pre-loaded with solvent (hexane (4): ethyl ether (1)) and left to run until the solvent could be visualised 1cm prior to the edge of the plate. Plates were then removed from the chamber, allowed to dry, and stained with CAM solution. Plates were baked at 80˚C for 15-25min and imaged using a scanner. Analysis was conducted in ImageJ using the Gels Analysis tool. This assay was repeated at least twice with four samples/treatment/experiment.

## Measurement of carbohydrates (glucose + trehalose and glycogen)

Each sample contained three flies that were homogenised in 75μl of TE + 0.1% Triton X-100 (Sigma Aldrich). Samples were incubated for 20 min at 75˚C and stored at −80˚C. Prior to the assay, samples were incubated for 5 min at 65˚C. Following incubation, 10μl from each sample was loaded into four wells of a 96-well plate. Each well was designated to serve as a measurement for either: control (10μl sample + 190μl H$_2$O), glucose (10μl sample + 190μl glucose reagent (Sentinel Diagnostics)), trehalose (10μl sample + 190μl glucose reagent + trehalase (Sigma Aldrich)), or glycogen (10μl sample + 190μl glucose reagent + amyloglucosidase (Sigma Aldrich)). A standard curve was generated by serially diluting a glucose sample of known concentration and adding 190μl of glucose reagent to 10μl of each standard. Standards were always run at the same time and in the same plate as samples. Plates were incubated for 1h at 37˚C following which the absorbance for each well at 492 nm was determined using a plate reader. This assay was repeated at least twice with four samples/treatment/experiment.

## Western Blots

Each sample contained three flies that were homogenised directly in 75μl 2x Laemmli SDS-PAGE buffer. The primary antibodies used were anti-phospho-Akt (Cell Signalling Technologies 4054, used at 1:1000), anti-total-Akt (Cell Signalling Technologies 4691, 1:1000), and anti-α-tubulin (Developmental Studies Hybridoma Bank 12G10, 1:5000). The secondary antibodies used were anti-rabbit IgG (Cell Signalling Technologies 7074, 1:5000) and anti-mouse

IgG (Cell Signalling Technologies 7076, 1:10,000). The chemiluminescent substrate used was SuperSignal West Pico PLUS (Thermo Scientific 34580). Blots were imaged using a Fuji LAS-3000 luminescent image analyser and images analysed in ImageJ.

## Feeding

For each sample, eight flies were placed into a 50mL Falcon tube with a lid containing our standard food (described above) with the addition of the food dye Erioglaucine sodium salt, 1% w/vol (Alfa Aesar) and left for 30 or 80h. To determine the amount of ingested food, flies were homogenised in 200µL of Tris-EDTA 0.1% Triton-X. Following homogenisation samples were spun down (20 minutes at 13 000 rpm, RT) and 100µL of the supernatant removed (this contained predominantly suspended triglyceride). We then added 300µL of Tris-EDTA 0.1% Triton-X to the sample and spun for 10 minutes at 13 000 rpm, RT. 200µL of each suspension was placed into a 96-well plate. We included a carcass only sample to control for differences in absorbance due to the unique constitution of individual flies. To determine the amount of excreted food, 1mL of Tris-EDTA 0.1% Triton-X was added to each Falcon tube and the tube briefly vortexed. Following vortex, tubes were placed on a roller for 5 minutes and then subject to a 'quick-spin'. 200µL was taken from each tube and placed into a 96-well plate. 96-well plates were read at 620nm and normalized to the mean value of the uninfected controls. Data are presented as a combination of excreted (Falcon tube) and ingested (fly homogenate) values. This assay was repeated at least twice with four samples/treatment/experiment.

## Statistical analysis

Data were analysed in R Studio with R versions 3.5.3 and 3.6.3 [94]. Behavioural data were analysed using the Rethomics package [41]; for all other assays, we first tested for normality of data which dictated whether an ANOVA, t-test, Kruskal-Wallis analysis of variance, or Mann-Whitney U test was used to calculate differences between treatments. When appropriate, we performed *post hoc* Tukey, Nemenyi or Dunn analyses to identify specific differences between treatments. All assays were repeated at least twice with sample sizes as indicated within the reported statistics.

## Supporting information

**S1 Fig. *Drosophila melanogaster* females do not show increased activity during infection.** Ethogram showing percentage of time wild-type female flies spend moving over time in 30-min bins during infection with (**A**) *Francisella novicida* or (**B**) *Micrococcus luteus* (**F. novicida**: Kruskal-Wallis chi-square = 1.042, df = 2, n = 240, p = 0.594; **M. luteus**: Kruskal-Wallis chi-square = 3.263, df = 2, n = 220, p = 0.196).
(PDF)

**S2 Fig. Quantifying engagement in specific behaviours.** Ethogram showing percentage of time infected wild-type males spend (**A**) engaging in micromovements (e.g. feeding and grooming), (**B**) walking and (**C**) sleeping in 30-min bins. Boxplots show the quantification of (**D**) total distance covered when flies were scored as awake and (**E**) the total number of times that the flies crossed the middle of the housing tube as a proportion of the time (in seconds) spent awake. Uninfected and mock controls are represented by grey and black tracings, respectively. Infected flies are in blue. **Distance covered normalized to time awake** (Kruskal-Wallis chi-square = 6.496, df = 2, n = 419, p = 0.039; Dunn's *post hoc*: mock|*F. novicida* = 0.033, mock|uninfected = 0.163, uninfected|*F. novicida* = 0.460) and **midline crosses normalized to time awake** (Kruskal-Wallis chi-square = 5.453, df = 2, n = 419, p = 0.065; Dunn's *post hoc*: mock|*F.

*novicida* = 0.064, mock |uninfected = 0.330, uninfected|*F. novicida* = 0.339) were not impacted by the infection.
(PDF)

**S3 Fig. Temporal comparison of activity during infection with *F. novicida*. (A)** Boxplots showing day-by-day activity of infected wild-type males. **12-36h**: (Kruskal-Wallis chi-square = 14.085, df = 2, n = 419, p = 8.7e-04; Dunn's *post hoc*: mock|*F. novicida* = 0.001, mock |uninfected = 0.346, uninfected|*F. novicida* = 0.02). **36-60h**: (Kruskal-Wallis chi-square = 98.325, df = 2, n = 419, p = 2.2e-16; Dunn's *post hoc*: mock|*F. novicida* = 3.1e-17, mock |uninfected = 0.519, uninfected|*F. novicida* = 7.9e-15). **60-84h**: (Kruskal-Wallis chi-square = 137.88, df = 2, n = 419, p = 2.2e-16; Dunn's *post hoc*: mock|*F. novicida* = 1.3e-22, mock |uninfected = 0.869, uninfected|*F. novicida* = 9.9e-22). **(B)** Activity level throughout infection was not correlated with survival (Pearson's correlation, $r$ = 0.313; t = 3.31, df = 101, p = 0.001) and **(C)** activity levels on day 1 are positively correlated with total activity (Pearson's correlation, $r$ = 0.744; t = 11.2, df = 101, p = 2.2e-16). Data from multiple replicates are shown.
(PDF)

**S4 Fig. Not all bacteria induce activity in wild-type flies.** Ethograms showing percentage of time flies spend moving over time in 30-min bins. Uninfected and mock controls are represented by grey and black tracings, respectively. Infected flies are in orange. Neither of these extracellular/Gram-negative bacteria, *Escherichia coli* and *Enterobacter cloacae* induced activity (**E. coli**: Kruskal-Wallis chi-square = 3.699, df = 2, n = 300, p = 0.16; **E. cloacae**: Kruskal-Wallis chi-square = 1.516, df = 2, n = 229, p = 0.47). Data from multiple replicates are shown.
(PDF)

**S5 Fig. *Francisella novicida* infection increases activity in several immune and locomotor mutants.** Boxplots showing the percentage of time flies spend moving over time. Uninfected and mock controls are represented by grey and black tracings, respectively. Infected flies are in blue. Previously characterized phenotypes and statistics of studied mutants can be found in S1 and S2 Tables. Data from multiple replicates are shown.
(PDF)

**S6 Fig. Bacterial growth and virulence. (A–J)** In all plots, grey and black tracings represent uninfected and mock controls, respectively. *F. novicida*, *Micrococcus luteus*, *Listeria monocytogenes* and *Staphylococcus aureus* infections shown in blue, green, orange and yellow, respectively. *Francisella novicida* infection was lethal in all four genotypes. Infection with *M. luteus* did not result in more lethality than either uninfected or mock controls, while *L. monocytogenes* and *S. aureus* both lead to decreased survival. Median survival is indicated by dotted lines intersecting the y and x axes at 50% survival and time (in days), respectively. Survival was calculated at the same time as activity data and thus have the same sample size as indicated elsewhere. Data from multiple replicates are shown. **(K)** Quantification of *M. luteus* markers represent means and whiskers represent SE. Initial inoculum consisted of ~ 5000 colony forming units (CFUs). Within 30h bacterial numbers decreased to near-undetectable levels (average of 28–40 CFUs/fly). **(L)** Quantification of *F. novicida*. Bacterial numbers increase over the course of infection. All genotypes were injected with the same initial dose (t = 0; ~1700 CFUs). The last measured timepoint was 24h prior to the onset of death for each genotype; this was 72h for all genotypes except *imd*[10191] which was 48h. Genotypes are represented by marker style and line colour as indicated inset. Markers indicate means and whiskers represent SE. Bacterial quantifications were repeated at least twice, n = 16–22 flies/genotype/timepoint; data from all replicates are shown.
(PDF)

**S7 Fig. The increase in activity during *S. aureus* infection is dependent on the Toll pathway.** Ethogram showing percentage of time $spz^{\Delta 8-1}$ mutant flies spend moving over time in 30-min bins after infection with *Staphylococcus aureus* (Kruskal-Wallis chi-square = 9.276, df = 2, n = 219, p = 9.68e-03; Dunn's *post hoc*: mock|*S. aureus* = 0.121, mock|uninfected = 0.221, uninfected|*S. aureus* = 7.56e-03).
(PDF)

**S8 Fig. Mutants of Toll and IMD pathways exhibit increased activity during infection.** Ethogram showing percentage of time **(A)** $imd^{10191}$ **(B)** $spz^{eGFP}$ and **(C)** $imd^{10191}$; $spz^{eGFP}$, flies spend moving in 30-min bins during *Francisella novicida* infection. Infected animals moved significantly more than both the uninfected and mock controls (**imd**: Kruskal-Wallis chi-square = 111.32, df = 2, n = 482, p = 2.2e-16; Dunn's *post hoc*: mock|*F. novicida* = 1.3e-20, mock|uninfected = 0.38, uninfected|*F. novicida* = 1.4e-18; $\boldsymbol{spz}^{eGFP}$: Kruskal-Wallis chi-square = 59.59, df = 2, n = 220, p = 1.1e-13; Dunn's *post hoc*: mock|*F. novicida* = 1.7e-12, mock|uninfected = 0.36, uninfected|*F. novicida* = 1.6e-09; $\boldsymbol{imd;spz}^{eGFP}$: Kruskal-Wallis chi-square = 52.594, df = 2, n = 195, p = 3.8e-12; Dunn's *post hoc*: mock|*F. novicida* = 5.6e-09, mock|uninfected = 0.45, uninfected|*F. novicida* = 1.9e-10). Data from multiple replicates are shown. **(D)** $imd^{10191}$ mutant flies exhibit increased activity during infection with *Listeria monocytogenes* (Kruskal-Wallis chi-square = 35.306, df = 2, n = 223, p = 2.16e-08; Dunn's *post hoc*: mock|*L. monocytogenes* = 1.22e-07, mock|uninfected = 0.454, uninfected| *L. monocytogenes* = 3.13e-06).
(PDF)

**S9 Fig. Injections with attenuated bacteria.** Ethogram showing percentage of time wild-type flies spend moving over time in 30-min bins during infection with **(A)** heat killed *Francisella novicida*, **(B)** heat killed *Micrococcus luteus* or **(C)** a low dose (~170 CFUs) of *F. novicida*. Injection with heat-killed *F. novicida* did not result in a significant increase in activity (Kruskal-Wallis chi-square = 18.40, df = 2, n = 240, p = 1e-04; Dunn's *post hoc*: mock|HK *F. novicida* = 0.011, mock|uninfected = 6.70e-05, uninfected|HK *F. novicida* = 0.120). Flies injected with heat-killed *M. luteus* moved significantly more than both the uninfected and mock controls (Kruskal-Wallis chi-square = 21.66, df = 2, n = 222, p = 1.98e-05; Dunn's *post hoc*: mock| HK *M. luteus* = 9.89e-06, mock|uninfected = 1.09e-02, uninfected|HK *M. luteus* = 4.16e-02). Infection with low dose of *F. novicida* led to increased activity (Kruskal-Wallis chi-square = 37.315, df = 2, n = 240, p = 7.89e-09; Dunn's *post hoc*: mock|low-*F. novicida* = 1.95e-07, mock|uninfected = 0.991, uninfected| low- *F. novicida* = 3.05e-07).
(PDF)

**S10 Fig. Persistent, relatively low levels of non-proliferating *F. novicida* does not induce activity.** Ethogram showing percentage of time wild-type male flies spend moving over time in 30-min bins during infection with **(A)** wild-type *Francisella novicida* treated with the antibiotic tetracycline, **(B)** shows an inset of (A) highlighting days 10–12 after infection (Kruskal-Wallis chi-square = 4.138, df = 2, n = 334, p = 0.126). **(C)** Ethogram for wild-type flies infected with a tetracycline-resistant strain of *F. novicida* under tetracycline treatment (Kruskal-Wallis chi-square = 65.011, df = 2, n = 274, p = 7.64e-15; Dunn's *post hoc*: mock|tet-R *F. novicida* = 2.18e-10, mock|uninfected = 0.196, uninfected| tet-R *F. novicida* = 1.88e-13). **(D)** Survival plot for flies treated with tetracycline and infected with wild-type and tetR *F. novicida*.
(PDF)

**S11 Fig. Pan-neuronal knock-down of the adipokinetic hormone receptor (AKH-R) inhibits the increase of activity induced by starvation.** Ethogram showing percentage of time **(A)**

AkhR-IR>+, **(B)** n-sybG4>+, and **(C)** n-sybG4/UAS—AKH-R RNAi male flies spend moving over time in 30-min bins. The grey area indicates the starvation period, when flies were transferred from standard food to starvation food (2% agarose in PBS). Boxplots indicate the quantification of the entire starvation period (Kruskal-Wallis chi-square = 39.088, df = 5, n = 231, p = 2.28e-07; Dunn's *post hoc*: UAS—AKH-R RNAi>+ FED|STARVED = 2.33e-05, n-sybG4>+ FED|STARVED = 2.80e-02, n-syb>AKH-R RNAi = 0.480). All genotypes were assayed at the same time but are here shown separately for clarity. Experiments conducted at 25˚C.
(PDF)

**S1 Table. Phenotypes of activity mutants tested.** Reported phenotypes for the tested activity mutants associated with immune response or general physical activity levels.
(DOCX)

**S2 Table. Statistics from activity mutant assays.** Statistics for differences in activity levels between infection conditions for the different mutants tested.
(DOCX)

**S3 Table. Fly strains.** *Drosophila melanogaster* strains used in this study and their sources.
(DOCX)

## Acknowledgments

We are indebted to members of the Dionne and Gilestro labs for critical discussion. Q. Geissmann provided invaluable feedback and support in the behavioural quantification.

## Author Contributions

**Conceptualization:** Crystal M. Vincent, Esteban J. Beckwith, Katrin Kierdorf, Giorgio F. Gilestro, Marc S. Dionne.

**Data curation:** Esteban J. Beckwith.

**Formal analysis:** Crystal M. Vincent, Esteban J. Beckwith.

**Funding acquisition:** Crystal M. Vincent, Giorgio F. Gilestro, Marc S. Dionne.

**Investigation:** Crystal M. Vincent, Esteban J. Beckwith, Carolina J. Simoes da Silva, William H. Pearson, Katrin Kierdorf.

**Methodology:** Giorgio F. Gilestro.

**Project administration:** Marc S. Dionne.

**Supervision:** Marc S. Dionne.

**Visualization:** Esteban J. Beckwith.

**Writing – original draft:** Crystal M. Vincent.

**Writing – review & editing:** Crystal M. Vincent, Esteban J. Beckwith, Carolina J. Simoes da Silva, William H. Pearson, Katrin Kierdorf, Giorgio F. Gilestro, Marc S. Dionne.

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
