## [Decision Letter · Decision Letter 0]

17 Mar 2022

Dear Dr. Dionne,

Thank you very much for submitting your manuscript "Infection increases activity via Toll dependent and independent mechanisms in Drosophila melanogaster" for consideration at PLOS Pathogens. As with all papers reviewed by the journal, your manuscript was reviewed by members of the editorial board and by several independent reviewers. In light of the reviews (below this email), we would like to invite the resubmission of a significantly-revised version that takes into account the reviewers' comments.

For a resubmission, you should consider the comments of reviewers 2 and 3, some of them are already suggested in the planed revisions:

1. Repeating the analyses in at least one additional genetic background (reviewer 3)..

2. Challenging the effect of dead bacteria (reviewer 2 and 3).

3. For the key bacterial strains, (F novicida and M luteus), determining whether hyperactivity is induced in both males and females (reviewer 3).

4. For the key bacterial strains, determining the minimum OD (reviewer 2).

5. Working at the minimal OD, may be helpful to clarify the claim based on results presented in Fig. 5; as stated by reviewer 3, these data are not convincing . If more significant effects can be obtained in this way, controls corresponding to Fig. S7 should be included in the new Fig. 5.

6. Working at the minimal OD, may also be helpful to clarify the variabilities observed in the mutant analysis (Fig. S4); in particular the apparent stronger effect for upd2 and dop1R1 mutants.

7. Challenging a potential effect of glial cells (reviewer 2).

8. For ease of the reading, modifying the presentation of some results (reviewer 3).

We cannot make any decision about publication until we have seen the revised manuscript and your response to the reviewers' comments. Your revised manuscript is also likely to be sent to reviewers for further evaluation.

Sincerely,

Jacques Montagne

Guest Editor

PLOS Pathogens

Guy TRAN VAN NHIEU

Section Editor

PLOS Pathogens

Kasturi Haldar

Editor-in-Chief

PLOS Pathogens

orcid.org/0000-0001-5065-158X

Michael Malim

Editor-in-Chief

PLOS Pathogens

orcid.org/0000-0002-7699-2064

For a resubmission, you should consider the comments of reviewers 2 and 3, some of them are already suggested in the planed revisions:

1. Repeating the analyses in at least one additional genetic background (reviewer 3)..

2. Challenging the effect of dead bacteria (reviewer 2 and 3).

3. For the key bacterial strains, (F novicida and M luteus), determining whether hyperactivity is induced in both males and females (reviewer 3).

4. For the key bacterial strains, determining the minimum OD (reviewer 2).

5. Working at the minimal OD, may be helpful to clarify the claim based on results presented in Fig. 5; as stated by reviewer 3, these data are not convincing . If more significant effects can be obtained in this way, controls corresponding to Fig. S7 should be included in the new Fig. 5.

6. Working at the minimal OD, may also be helpful to clarify the variabilities observed in the mutant analysis (Fig. S4); in particular the apparent stronger effect for upd2 and dop1R1 mutants.

7. Challenging a potential effect of glial cells (reviewer 2).

8. For ease of the reading, modifying the presentation of some results (reviewer 3).

Reviewer's Responses to Questions

**Part I - Summary**

Reviewer #1: This is an all-around excellent manuscript.

The authors demonstrate that increased activity is a common outcome from bacterial infections in the fly. Given two pattern recognition pathways, they trace these to determine which is responsible for this phenotype and come up with a complicated answer. That is OK, because it reflects the real biology of the host. I felt the authors did a good and fair job in discussing the previous literature on sleep/activity and infection in Drosophila. There aren’t a lot of papers on this but what exists seems contradictory. I agree with the author’s assessment that this apparent contradiction comes from differences in experimental and analysis and that this current manuscript provides a readily usable method of looking at infection.

I think a host’s response to infection goes far beyond the circle we draw around “immunity”. If hosts respond in a conserved way to infection, it seems like we should study it even if it falls outside of the Toll and imd signaling pathways in Drosophila. I’m particularly intrigued at the effects of signaling the authors show in the nervous system and look forward to seeing that work develop.

Reviewer #2: In this manuscript, Vincent et al, present their results regarding the effects of bacterial infection on fly behavior. To do so, they infect flies at a given time of the day and with different species of bacteria and measure the activity of the flies using the ethosccope behavioral platform.

Infection with the intra- and extracellular bacterium Francisella novicida shows that:

- Infected flies exhibit increased activity that intensifies during infection.

- This increase in activity is due to an increase in walking time and a reduction in sleeping time.

- Activity on the first day of infection is predictive of lifespan, with more active flies having a longer lifespan.

When the study was conducted with a larger number of species, the findings were different.

- Only 3 of the 5 species tested show an increase in activity.

- The correlation between increased activity and lifespan is not observed with all species.

- The increased activity is not affected in immune, locomotor and circadian mutants.

The authors then focus on infection with Francisella novicida and Micrococcus luteus.

To further characterize the observed effects, they analyze the metabolic changes upon infection. They confirm previous work showing that bacterial infection can lead to hyperglycemia and loss of triglyceride and glycogen stores and demonstrate that the increase in activity after infection is not a response to starvation. Finally, focusing on M. luteus infection, they show that fat body spz and neuronal Toll contribute to the activity induced by M. luteus infection.

This study demonstrates that the effects of bacterial infection on behavior are truly bacteria-specific, some do and some do not. The mechanism underlying the changes in fly activity is species specific. This result is well documented and demonstrated but, I suppose, it was expected. Since the species used are very different (Gram + or Gram -, pathogenic or not, immune cleared or not, intra or extra cellular...) I do not find it surprising that they influence fly activities differently. The second part is more focused on the mechanisms and I find interesting the result showing that the Toll pathway is involved.

Here are some suggestions for improving the manuscript.

- For some of the species, the authors should perform a dose response with different concentrations of bacteria. This should allow us to know if the observed effects are concentration-dependent. It would have been better, I think, to work on the minimum OD that gives an effect. Doses that are higher could mask some effects or show additive effects.

- The authors should compare dead and live bacteria as a trigger.

- I am surprised by the variability of the effects observed, with some differences being barely statistically significant and others showing a very strong effect. Is there an explanation?

- The authors should further characterize the Spz /Toll axis they identify. First, they only tested the effects of inactivating members of the Toll pathway in neurons. Glial cells are also known to respond to bacteria. Are they necessary here too? Are the Toll pathway members also needed in glial cells and if so, in which ones (perineurial, ensheating glia…?).

- The authors should also remove the Toll receptor itself from neurons and glial cells and test whether this impact the flies response to bacteria.

- Similarly, Toll pathway-dependent AMP production following M. luteus activation is mediated by pattern recognition receptors such as PGRP-SA and GNBP-1. Are these involved here? These experiments are important because they will tell us whether a completely different mode of Toll pathway activation is at play here.

- It is known that some fat body cells are present around the head. Can the authors find out if the spz ligand that is required to modulate fly activity after infection is produced from both the body and the fat body of the head?

- Hemocytes have been shown to mediate many interorgan communications in flies. To test whether they are also involved here, the authors should monitor the behavior of infected flies in which hemocytes have been removed.

Reviewer #3: Please refer to the attached letter

**Part II – Major Issues: Key Experiments Required for Acceptance**

Reviewer #1: (No Response)

Reviewer #2: (No Response)

Reviewer #3: Please refer to the attached letter

**Part III – Minor Issues: Editorial and Data Presentation Modifications**

Reviewer #1: I’ve not worked with an ms that came through review commons before. I agree with the author’s response to the reviewers. That the authors see a phenotype in different genetic backgrounds (even if the strength of that phenotype varies) strengthens their argument and does not require that everything be back-crossed into a single background. The phenotype is robust and that is a good thing.

I agree that it is OK to leave out the cellular immune response. One doesn’t have to do every possible experiment a reviewer can conceive of.

I certainly agree that it is ridiculous to ask the authors to now do a genetic screen. The authors are right in that would expose them to jeopardy as they would then need to figure out a molecular mechanism because that is what this sort of reviewer would ask for next. The paper is excellent as it stands. The one regret I have with dealing with review commons is that I can’t send a message to that reviewer to accept what they have in front of them and not ask for several years’ worth of work.

Reviewer #2: (No Response)

Reviewer #3: Please refer to the attached letter

PLOS authors have the option to publish the peer review history of their article (what does this mean?). If published, this will include your full peer review and any attached files.

Reviewer #1: No

Reviewer #2: No

Reviewer #3: No
---

## [Decision Letter · Decision Letter 1]

23 Aug 2022

Dear Dr. Dionne,

We are pleased to inform you that your manuscript 'Infection increases activity via Toll dependent and independent mechanisms in Drosophila melanogaster' has been provisionally accepted for publication in PLOS Pathogens.

Before your manuscript can be formally accepted you will need to make minor modfications in the text according to the comments of reviewer-3. You will also need to complete some formatting changes, which you will receive in a follow up email. A member of our team will be in touch with a set of requests.

Best regards,

Jacques Montagne

Guest Editor

PLOS Pathogens

Guy TRAN VAN NHIEU

Section Editor

PLOS Pathogens

Kasturi Haldar

Editor-in-Chief

PLOS Pathogens

orcid.org/0000-0001-5065-158X

Michael Malim

Editor-in-Chief

PLOS Pathogens

orcid.org/0000-0002-7699-2064

Please, follow reviewer-3 suggestions for minor modifications of the text that will improve the quality of your manuscript.

Reviewer Comments (if any, and for reference):

Reviewer's Responses to Questions

**Part I - Summary**

Reviewer #2: (No Response)

Reviewer #3: Together, the data presented in this revised manuscript show that infection induces activity in several Drosophila control lines. This effect is sex-dependent and associated with an increase in survival during infection. Increased locomotor activity could therefore play protective roles in infected host, and the data suggest that it could exert these effects, at least in parts, by promoting the acquisition of nutritional resources. This study also supports the existence of several bacterial factors promoting activity in male Drosophila: while Gram positive bacteria could act trough the release of peptidoglycan fragments and host bacterial sensing to promote activity, Gram negative bacteria employ different mechanisms and need to be metabolically active to induce these effects.

Several research groups previously described an opposite effect for bacterial infection on Drosophila sleep. While the reasons for these discrepancies remain elusive, it is possible that Vincent and colleagues made different observations because they studied males, or used activity monitors that are more sensitive and/or more accurate than the devices used in previous studies. I am convinced that solid observations with clear experimental procedures can only benefit to our understanding of these differences, and more importantly to our understanding of how bacteria regulate host behavior. This study by Vincent and colleagues contributes to this effort.

Overall, the authors answered to most of my comments. I still have, however, reserves regarding two of the conclusions presented in this revised manuscript, which I detail below.

**Part II – Major Issues: Key Experiments Required for Acceptance**

Reviewer #2: (No Response)

Reviewer #3: I am convinced that fat-body derived spz is required for inducing activity during M.luteus and S. aureus infection (Fig. 3B, 3C, 5A). The data, however, does not support a role for downstream Toll signaling in the fat body: the flies in which Myd88 or dif is silenced in the adipose tissue still show increased activity when infected with M. luteus (Fig. 5B-C: these flies show a significant increase in activity when compared to mock controls, particularly visible on the ethogram in Figure 5B). Therefore, the authors cannot claim that “increased activity after M. luteus infection requires Toll pathway activity in the fat body” in the abstract (ln. 32), results (ln. 263-65), and discussion (ln. 374-376). In contrast, the possibility that “fat body derived spz activates Toll signaling in other tissues” to promote activity can still be considered, until proven wrong (Discussion, ln 378-79).

In fact, there is no data unambiguously showing that Toll signaling downstream of Spz plays a role in infection-induced activity (Do Myd88 mutants behave as Spz mutants when infected with M. luteus and S. aureus?). I think it is therefore safer to not make strong statements about the tissue-specific roles for Toll signaling in promoting activity during infection in the manuscript (including the abstract). Rather, these points should be mentioned as likely “hypotheses” in the discussion, unless additional data is provided to support these claims.

Finally, there are issues with the interpretation of the neuronal experiments (Fig. 5E-G). Based on their results, the authors conclude that “Neuronal KD of Toll signaling does not affect activity during M. luteus infection” (Ln 267).

Yet, a few lines later, they state the opposite (we “demonstrate that neuronal Toll signaling plays a role in infection-induced changes in activity” (ln. 288)). The authors further propose a model whereby fat body-derived spz acts in neurons to induce toll and promote activity during infection, which is invalidated by their data (ln. 372-73). These parts must be corrected.

**Part III – Minor Issues: Editorial and Data Presentation Modifications**

Reviewer #2: (No Response)

Reviewer #3: It could be good to mention earlier in the results which bacterial infections are lethal (ln 116 rather than 134?). This would bring clarity, and justify why not looking at the correlation between activity and survival during M. luteus infection (since it does not kill), or why scoring activity only during the first day after S. aureus infection (since flies succumb rapidly under this condition).

Ln 172: Maybe mention the reference to Figure S6 K, L earlier in the text?

Ln 361: Infection-induceD

Ln 394: “control” rather than “wild-type”?

PLOS authors have the option to publish the peer review history of their article (what does this mean?). If published, this will include your full peer review and any attached files.

Reviewer #2: No

Reviewer #3: No

---

## [Editor Report · Acceptance letter]

16 Sep 2022

Dear Dr. Dionne,

We are delighted to inform you that your manuscript, "Infection increases activity via Toll dependent and independent mechanisms in *Drosophila melanogaster*," has been formally accepted for publication in PLOS Pathogens.

Best regards,

Kasturi Haldar

Editor-in-Chief

PLOS Pathogens

orcid.org/0000-0001-5065-158X

Michael Malim

Editor-in-Chief

PLOS Pathogens

orcid.org/0000-0002-7699-2064